# Assessing Implantation Sites for Pancreatic Islet Cell Transplantation: Implications for Type 1 Diabetes Mellitus Treatment

**DOI:** 10.3390/bioengineering12050499

**Published:** 2025-05-09

**Authors:** Vinícius Gabriel Silvério Scholl, Leonardo Todeschini Justus, Otávio Simões Girotto, Kelly Karine Pasqual, Matheus Henrique Herminio Garcia, Fernando Gonçalves da Silva Petronio, Aline Flores de Moraes, Sandra Maria Barbalho, Adriano Cressoni Araújo, Lucas Fornari Laurindo, Cristina Pires Camargo, Maria Angélica Miglino

**Affiliations:** 1Department of Biochemistry and Pharmacology, School of Medicine, Universidade de Marília (UNIMAR), Marília 17525-902, SP, Brazil; vinicius.scholl@hotmail.com (V.G.S.S.); ltjustus@hotmail.com (L.T.J.); otgirotto@gmail.com (O.S.G.); kellypasqual@gmail.com (K.K.P.); fernandogspetronio@gmail.com (F.G.d.S.P.); alinefloresmoraes@hotmail.com (A.F.d.M.); smbarbalho@gmail.com (S.M.B.); adrianocressoniaraujo@yahoo.com.br (A.C.A.);; 2Postgraduate Program in Animal Health, Production and Environment, School of Veterinary Medicine, Universidade de Marília (UNIMAR), Marília 17525-902, SP, Brazil; matheushenrigarcia@gmail.com; 3Postgraduate Program in Structural and Functional Interactions in Rehabilitation, School of Medicine, Universidade de Marília (UNIMAR), Marília 17525-902, SP, Brazil; 4Department of Biochemistry and Nutrition, School of Food and Technology of Marília (FATEC), Marília 17500-000, SP, Brazil; 5Microsurgery and Plastic Surgery Laboratory (LIM-04), Faculdade de Medicina, Universidade de São Paulo, São Paulo 05508-220, SP, Brazil; cristinacamargo@usp.br; 6Department of Animal Anatomy, School of Veterinary Medicine, Universidade de Marília (UNIMAR), Marília 17525-902, SP, Brazil

**Keywords:** pancreatic islet transplantation, diabetes mellitus, artificial pancreas, decellularized pancreas, recellularized pancreas

## Abstract

Type 1 diabetes mellitus (T1DM) involves the destruction of pancreatic β-cells, requiring ongoing insulin therapy. A promising alternative for management is pancreatic islet transplantation, or the bioartificial pancreas. Here, we examine the primary implantation sites for the bioartificial pancreas, highlighting their anatomical, physical, and immunological characteristics in the context of T1DM treatment. Traditionally used for islet transplantation, the liver promotes metabolic efficiency due to portal drainage; however, it presents issues such as hypoxia and inflammatory responses. The omentum offers excellent vascularization but has limited capacity for subsequent transplants. The renal subcapsular space is advantageous when combined with kidney transplants; however, its use is limited due to low vascularization. The subcutaneous space is notable for its accessibility and lower invasiveness, although its poor vascularization poses significant challenges. These challenges can be mitigated with bioengineering strategies. The gastrointestinal submucosa provides easy access and good vascularization, which makes it a promising option for endoscopic approaches. Additionally, the intrapleural space, which remains underexplored, offers benefits such as increased oxygenation and reduced inflammatory response. Selecting the ideal site for bioartificial pancreas implantation should balance graft support, complication reduction, and surgical accessibility. Bioengineered devices and scaffolds can address the limitations of traditional sites and enhance T1DM management.

## 1. Introduction

Diabetes mellitus affects approximately 387 million people worldwide, and the number of cases is projected to increase by 53%, reaching 592 million by 2035 [1,2,3]. Type 1 diabetes mellitus (T1DM), a disease that requires continuous insulin dependence, has increased significantly in various countries. The severe complications and mortality rates associated with T1DM are higher compared to those of type 2 diabetes mellitus (T2DM). Moreover, the intensive and continuous insulin treatment essential for T1DM patients imposes considerable physical, psychological, and economic challenges [4,5,6].

The management of T1DM has advanced due to innovations in glucose monitoring, the development of the artificial pancreas, and alternative routes for insulin delivery. Emerging technologies promise to enhance insulin administration by integrating precise glucose level measurements with delivery systems tailored to individual needs. Innovative delivery methods, such as needle-free technologies, hold the potential to optimize glycemic control, minimize invasiveness, and improve treatment adherence [7,8,9,10,11,12].

Pancreatic islets have demonstrated a superior ability to regulate blood glucose levels compared to current artificial pancreas systems. For this reason, pancreatic islet transplantation holds excellent potential for treating T1DM [13,14,15,16]. However, despite advances in indications for islet transplantation and autotransplantation, substantial challenges persist, including the need for immunosuppression and site-specific issues, such as hypoxia [17,18,19,20].

To ensure an adequate supply of nutrients, selecting a site where the artificial pancreas can be near the bloodstream is crucial. However, this selection is challenging because the site must be able to accommodate a significant graft volume. Thus, the ease of implantation, potential for graft retrieval, and proximity to nutrient sources are essential when choosing the implantation site [21,22,23].

Extravascular and intravascular implantation are two main options for bioartificial pancreas allocation [21]. Extravascular implantation in subcutaneous tissues offers advantages such as surgical practicality, ease of removal and replenishment, and reduced risk of damaging vital organs [24]. However, islet functionality may be compromised by a low surface-to-volume ratio, insufficient vascularization, and diffusion limitations. The geometry of extravascular devices and the avascular nature of the implantation site prevent the proper diffusion of nutrients and insulin due to large diffusion distances (>200 µm). This results in inadequate glucose–insulin kinetics, ultimately leading to the death of islet cells. Additionally, subcutaneous implantation triggers a rapid foreign-body response, characterized by the migration of macrophages to the implantation site, followed by their fusion into giant cells, eventually leading to fibrosis and an increase in the diffusion distance [24].

In this context, islet transplantation offers advantages over macrodevices. For example, microencapsulated islets have demonstrated adequate competence for insulin release [21]. However, islet function tends to deteriorate over time because islet isolation leads to the loss of vascularization and extracellular matrix proteins, culminating in decellularization [25]. On the other hand, devices such as decellularized pancreas scaffolds may be a viable alternative, as they preserve the tissue’s intrinsic vascularization. For this reason, alternative implantation sites, such as the spleen, kidney, omental pouch, gastric submucosa, small intestine, and vascularized subcutaneous devices, have also been explored, although with varied outcomes [26].

Exploring accessible implantation sites that allow modification of the local microenvironment is crucial for optimizing graft survival and functionality, reducing immune activation, and minimizing marginal mass loss during transplantation. However, no review has addressed the complete literature in one paper because many recent studies have been published over the past few years. For instance, Berney et al. [27] discussed the possible sites for bioartificial pancreas implantation in their paper. Nevertheless, their analysis delved more into the biology behind the microenvironment alterations for the success of the implant because their primary objective was not to assess the possibilities for implantation sites. In 2018, Zhu et al. [28] also reviewed the possible implantation sites for the bioartificial pancreas. Although their analysis has been cited very often, their paper lacks a comprehensive analysis of the recent findings, and there is no mention of the intrapleural space or the gastrointestinal submucosa in their article. Finally, although Hwang et al. [29] worked with implantation sites for the bioartificial pancreas in their article, their analyses delved even more into the microencapsulation techniques for bioartificial pancreas implantations because this was their primary objective, lacking comprehensive analyses of the implantation sites themselves and their pros and cons. The present review aims to identify and evaluate all potential sites for bioartificial pancreas transplantation. Our analysis considers critical factors, including the proximity to the bloodstream to provide the correct blood supply to the implant, graft support capacity to provide the most excellent environment to support the implant, and surgical feasibility, including the facility of the implantation sites to be reached and to accept the implants surgically. Our review also aims to discuss what implantation sites would best further T1DM treatment, as this is a crucial avenue for future research. No previous paper has addressed implantation sites for the bioartificial pancreas from this perspective. Therefore, this review aims to contribute to selecting the most suitable implantation site, thereby enhancing the functionality and longevity of the bioartificial pancreas in T1DM patients, with implications for future clinical practice.

## 2. Literature Search Methodology and Report

### 2.1. Materials and Methods

To identify and evaluate the main implantation sites for the bioartificial pancreas, we searched the PubMed/MEDLINE database using the keywords “bioartificial pancreas implantation site”, “pancreatic islets transplantation”, “pancreas tissue engineering”, and “type 1 diabetes treatment”. We deemed eligible studies that investigated the anatomical, histological, and functional characteristics of bioartificial pancreas implantation sites in human subjects or experimental models. The inclusion criteria were as follows: (1) studies on implantation sites for the bioartificial pancreas or pancreatic islets; (2) in vivo experimental or clinical studies; (3) articles published in English or Portuguese; and (4) full-text access. The exclusion criteria were: (1) literature reviews or meta-analyses, (2) studies not directly relevant to the topic, (3) duplicate studies, and (4) publications in languages other than English or Portuguese or without full-text access. We selected articles in two steps. The first step consisted of screening the titles and abstracts. The second step consisted of reviewing the full texts of eligible articles. Our evaluation focused on relevant information to compare different implantation sites, considering their advantages, limitations, and practical feasibility. All steps were conducted independently to ensure methodological rigor and the reliability of the results. The literature search was conducted between December 2024 and March 2025.

### 2.2. Search Report Following PRISMA Guidelines

Figure 1 depicts the literature search for the identification, screening, and final inclusion of the articles searched for in writing the present manuscript, following the Preferred Reporting Items for Systematic Reviews and Meta-Analyses (PRISMA) guidelines. Our literature search yielded 248 reports from PubMed following our initial identification stage. After further screening, 49 reports were excluded due to duplication, and an additional 98 records were marked as ineligible by automation tools. Twenty records were also excluded due to insufficient methodological rigor. During the screening phase, 81 records were initially screened. Thirty-seven were excluded because they were not written in the English language. At this stage, 44 reports were sought for retrieval. Fortunately, all 44 reports were successfully retrieved. However, seven reports were excluded when assessing these 44 for complete eligibility because they did not involve bioartificial pancreas transplantation. Ultimately, 37 reports were included. These were related to the liver (*n* = 9), omentum (*n* = 4), renal subcapsular (*n* = 5), subcutaneous space (*n* = 7), muscle (*n* = 4), gastrointestinal submucosa (*n* = 6), and intrapleural space (*n* = 2).

## 3. Challenges in Finding the Ideal Site for Bioartificial Pancreas Implantation

T1DM is characterized by the progressive destruction of insulin-producing pancreatic β-cells and affects individuals of all ages. Intensive insulin therapy can control blood glucose levels and reduce or prevent diabetes-related complications [1]. However, this method often results in recurrent episodes of severe hypoglycemia [30]. Moreover, patients with difficult-to-control diabetes or complications are usually challenging to treat. Alternatively, cell therapy, through the grafting of insulin-producing tissues or cells, offers a physiological alternative to insulin use, as demonstrated in islet transplantation studies [21].

Pancreatic islets typically have a rich blood supply, approximately ten times greater than the exocrine pancreas, like that observed in the renal cortex (approximately 5–7 mL × min^−1^ × g^−1^). This high blood flow is essential for supplying the islet cells with oxygen and nutrients, facilitating the dispersion of hormones to their target organs [25]. However, the isolation of pancreatic islets and subsequent in vitro culture disrupt the native vasculature, leading to cell dedifferentiation and degeneration. After transplantation, the islets rely on oxygen and nutrient diffusion from the surrounding tissues [25,31]. Revascularization generally occurs between 7 and 14 days, with complete revascularization achieved within one month post-transplantation [32].

One study found that one month after implantation, the vascular density of transplanted mouse pancreatic islets was significantly lower than that of endogenous islets [31]. This reduction in vascularization was observed at all assessed implantation sites but was more pronounced in islets transplanted into the spleen than in those transplanted into the liver or kidney. These findings suggest that the revascularization of transplanted islets is quantitatively compromised, regardless of the implantation site. Additionally, islets autotransplanted beneath the renal capsule of partially pancreatectomized animals showed blood flow more similar to that of endogenous islets [31].

Another study found that pancreatic islets transplanted under the renal capsule had significantly lower oxygenation levels than native islets, even after nine months [25]. Moreover, persistent hyperglycemia in diabetic recipients further reduced graft oxygenation, which suggests insufficient oxygenation in transplanted islets [25].

Early islet loss may be caused by inflammatory processes, the activation of blood coagulation, shear forces from portal blood flow, and inadequate extracellular matrix for grafting and islet reorganization [33]. Consequently, transplanting excessive islets is often necessary to achieve insulin independence. However, this approach can trigger adverse effects, such as the release of tissue factor, which activates platelets, granulocytes, and monocytes, ultimately leading to thrombosis, tissue injury, inflammation, and loss of transplanted islets [33].

The challenges of islet transplantation are significant and can inform studies to improve bioartificial pancreas transplantation. The search for the ideal site for bioartificial pancreas transplantation considers the ability to accommodate large graft volumes, portal drainage, ease of safe access, and reproducible procedures. Moreover, the implantation site must provide adequate blood and oxygen supply, minimize immune and inflammatory responses, be compatible with non-invasive imaging and biopsies, and allow for microenvironment manipulation or bioengineering interventions.

## 4. Bioartificial Pancreas Transplantation into the Liver

Insulin secreted by native β-cells is released into the portal venous system, where the liver and other tissues, such as muscles, primarily metabolize it. Therefore, it is reasonable to assume that the liver provides a physiologically suitable environment for islets, thus facilitating insulin effectiveness. Kemp et al. [34] were pioneers in this field. They used rat models to compare islet implantation into the portal vein with intraperitoneal implantation. Their findings indicated that diabetes remission occurred only when the islet transplantation was into the portal vein. Consequently, the liver became the primary target for islet transplantation studies and the most widely used site for intraportal injection. However, intravascular transplantation is hindered by an instant blood-mediated inflammatory reaction induced by tissue factors expressed on islet surfaces. This reaction leads to platelet activation and aggregation, initiation of the coagulation cascade, and increased infiltration of neutrophils, monocytes, and macrophages [35], significantly reducing the number of transplanted islets. This inflammatory response poses a challenge for all intravascular transplantation sites, restricting the space available for islets.

In one study involving seven diabetic patients, each received infusions of over 4000 pancreatic islets [11,547 ± 1604 islet equivalents (IEQ)] per kilogram of body weight, administered through the portal vein. All seven patients achieved insulin independence within the first year following the procedure [36]. However, the long-term outcomes of islet transplantation remain unsatisfactory, with only a few patients remaining insulin-independent five years post-transplantation [37]. Currently, it is common practice to perform multiple sequential transplants using pancreatic islets from two to three donors to treat a single patient with insulin-dependent diabetes mellitus [38].

Although the exact mechanisms underlying islet loss are not fully understood, the instant blood-mediated inflammatory reaction and the hypoxemic environment in the portal venous circulation are believed to play significant roles in this process [39]. After transplantation, the islets rely exclusively on nutrient and oxygen diffusion from the surrounding tissue. Therefore, islet size, graft composition, and oxygen levels in the recipient organ can strongly influence graft oxygenation.

Another study used the oxygen-dependent bioreductive metabolism of pimonidazole to detect low-oxygenated islets following intraportal transplantation [40]. The authors found that insufficient oxygenation and delayed revascularization increased hypoxic stress and apoptosis, particularly during the early post-transplantation period. Approximately 70% of the islets exhibited low oxygen levels (<10 mmHg), compromising graft function and survival. By contrast, subcapsular renal grafts exhibited better oxygenation and reduced apoptosis [40].

Pre-treating islets with anti-hypoxic agents before transplantation has shown potential for improving short-term islet survival; however, the need for extensive vascularization remains a significant challenge for long-term islet survival [41].

Intrahepatic transplantation has limitations concerning the volume of tissue that can be implanted, and it exposes the islets to potentially toxic concentrations of immunosuppressive agents in the portal circulation. Moreover, the use of the portal site is associated with severe complications, such as bleeding and portal vein thrombosis [42]. Additionally, portal islet transplantation may induce hepatocellular carcinoma [43].

Incompatibility of the major histocompatibility complex can trigger exaggerated immune responses, which makes the pancreatic islets’ origin a critical factor in their short- and long-term viability and functionality. When islets are transplanted from an incompatible donor, immunosuppressive regimens may be necessary to prevent or delay graft rejection, thereby prolonging their functional longevity [41]. In this context, decellularized pancreas transplants offer a promising alternative to reduce inflammation, minimize graft failure, and improve euglycemia while decreasing reliance on immunosuppressive medications that can harm pancreatic β-cell survival and function [44].

In this scenario, modifications to the liver microenvironment may challenge pancreatic cell survival within transplants [45,46]. Oxidative stress is linked to the progression of many diseases, including hepatic affections. Oxidation alters the liver microenvironment, causing diminished normal, healthy cell survival due to increased cellular toxicity and apoptosis [47]. Islet preparations supplemented with mesenchymal stromal cells may be an alternative to improve islet survival during the isolation and, possibly, transplantation processes, ultimately reducing islet stress and improving their functionality post-transplant [48]. Diminishing oxidative stress during islet transplantation may also contribute to long-term results in diabetic patients because the increased survival of transplants leads to better glucose homeostasis restoration and, therefore, lower later dysfunction [49].

Despite its prevalence in transplantation, the intraportal site has several limitations. A significant challenge is the progressive loss of islet function over time, as shown in studies of human and rodent transplants. These limitations have prompted research to improve islet oxygenation and vascularization, which are critical aspects for the success of intraportal transplantations.

The liver is a common alternative for islet cell transplantation because it has huge vascular access and can easily induce and support insulin production through implants [50,51]. However, in cases of pancreatic cancer requiring total pancreatectomy [52], the decision on the transplant location must consider several aspects, including the potential for the liver to receive pancreatic cancer metastasis [53], as this could impact the liver’s suitability for receiving implants. When the liver’s suitability is compromised, alternative sites must be considered, which are discussed in the following sections.

## 5. Bioartificial Pancreas Transplantation into the Omentum

The first clinical transplantation of pancreatic islets into the omentum occurred in the 1980s when three patients received allogeneic islet transplants into the epiploic arteriolar flap. Insulin production was observed in all cases. In one patient, islet transplantation was combined with liver transplantation. The outcome was successful, and the patient achieved insulin independence within seven months and maintained it for 15 months. This was one of the earliest examples of prolonged insulin withdrawal after islet transplantation [54].

The omentum is favorable for islet transplantation due to its excellent neovascularization capacity, regenerative properties, hemostatic characteristics, and immunologically privileged status. Located in the abdomen, the omentum primarily consists of an extensive layer of adipose tissue [55]. It facilitates physiological insulin drainage through the portal vein and avoids direct exposure to the bloodstream [56]. Since then, various approaches have been explored for islet transplantation into the omentum. One such approach involved wrapping a cell pouch device in the omentum, vascularizing it, and implanting pancreatic islets into the cell pouch [57]. Another study employed a h-omental matrix islet filling (hOMING) technique, which enabled inserting cells and hydrogel between the omental layers, near blood vessels, to maximize graft integration within a favorable metabolic environment [58]. In a study involving pancreatectomized dogs, the omental pouch improved the survival rate of unpurified islets compared to other sites [59]. The ability of the omentum to accommodate large volumes of low-purity islets makes it promising for allogeneic islet transplantation. This is particularly relevant given the low purity of isolated human islets and the difficulty of transplanting large volumes of unpurified islets into the liver [59,60].

Berkova et al. investigated the implantation of a decellularized pancreatic skeleton in the rat omentum. Because of the limited size of the omentum, only the pancreatic tail was used. Following laparotomy, the greater omentum was excised from the abdominal cavity and spread on a sterile field. A sterile extracellular matrix was then placed on the omentum, followed by the infusion of pancreatic islets. The omentum was subsequently pulled over the skeleton, and fixation was performed with a single stitch. Histological analyses and immunofluorescence imaging confirmed graft viability and sustained insulin production in non-diabetic syngeneic recipients [56].

Another study compared the omental and intrahepatic sites for allogeneic islet transplantation in diabetic mice [61]. The results showed that the minimum islet mass required to reverse diabetes in 50% of the mice, measured in IEQs, was lower in the omentum (200 IEQs) than in the liver (600 IEQs). Moreover, the average time to achieve euglycemia was shorter in the omentum transplant group (13.9 ± 3.7 days) than in the liver transplant group (15.1 ± 3.3 days). However, the omentum showed a significant limitation: it did not allow for repeated transplantation, which makes it unsuitable for patients requiring laparotomy for any reason. Additionally, accessing and manipulating the omentum carries a small risk of intestinal adhesion and obstruction.

## 6. Bioartificial Pancreas Transplantation into the Renal Subcapsular Space

Experimental islet transplantation under the renal capsule has been used by several researchers, yielding promising results. Studies in rats have shown that this technique not only normalizes blood glucose and insulin levels in response to various stimuli but also reverses serious diabetes-related complications, including diabetic nephropathy and retinopathy [62]. The renal subcapsular space has been proposed as an immunologically privileged site, which may contribute to the success of transplantation [63]. This approach also provides significant clinical benefits for combined islet and kidney transplants from the same donor, thus enabling islet implantation into the newly transplanted kidney.

Despite these benefits, the oxygen tension in islets transplanted under the renal capsule remains significantly lower than in native islets, even nine months post-transplantation [25]. This discrepancy arises because the blood supply to the subcapsular region is lower than that of the renal parenchyma and pancreatic tissue. In the renal subcapsular space, the oxygen tension is approximately 15 mmHg, markedly lower than the ~40 mmHg observed in the native environment of pancreatic islets. This reduced oxygen availability may compromise islet viability [64]. Another considerable challenge is that the renal capsule in large animals and humans is more rigid and restrictive than in rodents. This characteristic complicates the implantation of an adequate number of islets, increasing the risk of narrowing and compromising graft viability [65]. Furthermore, the procedure requires an invasive intervention (e.g., laparotomy), making it less appealing than other transplantation sites.

Although there are no clinical studies in humans, research has been conducted on large animals, such as pigs and non-human primates. A survey in primates highlighted the importance of prevascularization of islet–kidney grafts for successful transplantation. Nonvascularized islets experienced hypoxia and insufficient nutrient supply, whereas the islet–kidney model successfully achieved euglycemia using islets from a single organ [66]. In preclinical models with baboons and rhesus macaques, islet–kidney graft transplantation proved effective in achieving euglycemia and immune tolerance in patients with T1DM associated with chronic kidney disease. This technique reduces surgical complications and eliminates the need for multiple pancreases to reverse diabetes. In baboons, insulin independence was achieved using islets from a single pancreas. However, the efficacy was limited in rhesus macaques, highlighting the need for additional strategies, such as minimizing diabetogenic immunosuppressants and preserving islet function [66].

The islet–kidney approach emerges as a safe and effective alternative to simultaneous kidney–pancreas transplantation, with potential future clinical applications [67]. However, further research is needed for renal subcapsular transplantation to be clinically viable, including developing less invasive laparoscopic techniques and strategies to induce immune tolerance. Future efforts should also prioritize efficient donor screening and the development of noninvasive methods to assess islet function after transplantation.

## 7. Bioartificial Pancreas Transplantation into the Subcutaneous Space

The subcutaneous space is considered an attractive site for islet transplantation due to its accessibility through minimally invasive techniques. However, its limited neovascularization capacity leads to hypoxia and cell death, compromising graft longevity and functionality and consequently restricting its clinical application [68]. This limitation arises from the poor vascularization of subcutaneous tissue, which cannot support highly vascularized cells, such as pancreatic islets [69]. On the other hand, surgical injuries caused by scaffold insertion can trigger a metabolically active healing process in the skin and subcutaneous tissue [70]. Nevertheless, one study found that oxygen partial pressure in the subcutaneous space is higher than in the intraperitoneal and intramuscular spaces and is equivalent to that in hepatic and pancreatic tissues [68].

Halberstadt et al. [69] proposed creating a vascular bed in the subcutaneous space to precondition the site for islet cell transplantation. Their experiments demonstrated that transplanted islets generally responded to a glucose bolus and remained viable for up to 60 days with no signs of inflammation. However, only 73% of the animals that received the device became normoglycemic, similar to those that received renal subcapsular islet transplantation. No islet survival occurred in subcutaneous transplants without the device.

The CellSaic platform, which combines cells with petaloid pieces of recombinant peptide (RCP), has shown promising results by facilitating nutrient penetration, preventing cell death, and stimulating blood vessel formation in grafts [71]. CellSaic prevents central necrosis in cellular structures larger than 400 μm in diameter and offers improved cell viability compared to spheroids, both in vitro and in vivo [71]. These benefits arise from the interstitial spaces formed by RCP pieces, which aid in nutrient delivery and waste removal. Consequently, CellSaic enables the transplantation of larger cell tissues while preventing necrosis [71].

Another study reported that surgical scaffold insertion into subcutaneous tissue heals in four phases: (1) hemostasis, (2) wound inflammation, (3) vascularized granulation tissue formation, and (4) avascular healing [72]. Transplantation occurred in phase 3 with minimal stimuli. Subcutaneous transplantation of pancreatic islets at four islets per gram of recipient body weight (RBW) was less invasive and as effective as portal vein transplantation [72]. The proportions of alpha and beta cells suggest that artificially created subcutaneous cavities are viable for future islet transplantation therapies [72].

Yasunami et al. [38] developed a procedure for transplanting pancreatic islets into subcutaneous inguinal white adipose tissue (ISWAT) to form clusters. This method reversed streptozotocin-induced diabetes in mice more effectively than liver transplantation. In the ISWAT, 200 syngeneic islets (the amount obtained from a single mouse pancreas) controlled hyperglycemia, a result not achieved with liver transplantation. Human islets transplanted into the ISWAT also controlled hyperglycemia in streptozotocin-induced diabetic NOD/scid mice, with 2000–2500 IEQ doses.

Islet transplantation into unmodified subcutaneous sites has failed to reverse diabetes in animals or humans due to low oxygenation and insufficient vascularization [73]. However, methods that promote revascularization and angiogenesis could render subcutaneous tissue a viable and accessible site for islet transplantation.

## 8. Bioartificial Pancreas Transplantation into Muscle

The autotransplantation of parathyroid glands boosted the transplantation of pancreatic islets into muscle fibers. When implanted into forearm muscle fibers, parathyroid cells initiate the processes of grafting and revascularization. In the long term, this procedure yields excellent outcomes with minimal side effects [74]. Svensson et al. [75] found that the oxygen tension in two-month-old intramuscular islet grafts was six times higher than that observed in corresponding renal subcapsular grafts, reaching 70% of the levels seen in native islets.

The surgical procedure for intramuscular implantation is relatively straightforward. Additionally, it can be performed under local anesthesia with a low risk of complications. Pancreatic islets can also be transplanted into multiple muscle sites, thus enabling numerous tissue implantations and explantations, increasing the flexibility of this approach [68]. Although intramuscular pancreatic islet transplantation is a promising option for vascularization, it faces considerable challenges, such as a strong immune response and a reduced lifespan of allogeneic islets. Nevertheless, biocompatible materials and immunomodulatory strategies may mitigate these issues [76].

In a recent study, an electrospun polylactic-co-glycolic acid (PLGA) scaffold coated with polydopamine (PDA) was loaded with pancreatic islet cells and implanted into the skeletal muscle of rats with T1DM. One week after transplantation of the PDA-PLGA complex, blood glucose levels in the treated group were significantly lower than those in the model group (*p* < 0.001), and this difference persisted for approximately three weeks. However, allotransplantations are susceptible to immune rejection and mechanical pressure caused by skeletal muscle movement, which can ultimately lead to cell death [77].

In a clinical case, a 7-year-old girl with severe hereditary pancreatitis underwent a total pancreatectomy and received 160,000 islet cells into the brachioradialis muscle. Over two years, the patient exhibited normal glycated hemoglobin (HbA1c) levels, low insulin requirements, and no hypoglycemia episodes, which indicates that the transplanted islets contributed to effective long-term metabolic control [78]. In another study, four patients with T1DM received allogeneic human islet transplants into the brachioradialis muscle. Despite no surgical complications, the islets gradually lost functionality, highlighting the challenges associated with the intramuscular transplantation of allogeneic tissues [79].

Thus, islet transplantation into striated muscle is a feasible approach. However, the intramuscular site is currently less efficient due to a strong immune response and the short lifespan of allogeneic islets. However, manipulating the graft and implantation site before transplantation offers significant advantages for human applications.

## 9. Bioartificial Pancreas Transplantation into the Gastrointestinal Submucosa Space

In the 1970s, Champault et al. [80] conducted autotransplantations and allotransplantations of pancreatic fragments into the gastric submucosal space of rabbits. Following the transplantations, glucose levels remained within normal ranges for 60 days. In this context, the gastric submucosa has been considered a promising alternative site for biopancreatic transplantation due to its dense vascular network and ability to provide long-term trophic support in in vitro studies [81]. Moreover, studies have shown a reduced inflammatory response in various contexts. The gastric submucosa also offers easy access for grafting and subsequent evaluation, which enables its use via endoscopy in larger animal models and humans [82].

Wszola et al. [83] demonstrated that diabetic pigs endoscopically grafted with islets into the gastric submucosa exhibited improved glycemic control and a reduced need for insulin compared to the control group. Although previous studies had described grafting in the submucosal layer of the stomach in diabetic pigs, none had identified the ideal anatomical location for the graft, a point addressed in the study by the authors.

Caiazzo et al. [84] compared the implantation of pancreatic islets in the gastric submucosa and under the renal capsule of minipigs and observed better islet function in the gastric submucosa. They used an extracellular matrix rich in growth factors to improve vascularization and reduce graft necrosis. However, this technique was unsuccessful because the renal capsule could not accommodate the increased volume of the transplanted tissue. The results obtained in the gastric submucosa were also unsatisfactory, possibly because the Matrigel prevented the islets from interacting with the submucosal environment. By contrast, another study showed that islet autotransplantation in the liver significantly increased graft survival compared to transplantation in the gastric submucosa under the same conditions. These findings suggest that the liver may be a more suitable site than the gastric submucosa for islet transplantation in animal models [85].

The submucosal space of the duodenum was investigated in Syrian golden hamsters as a potential site for pancreatic islet transplantation. After transplantation, streptozotocin-induced diabetic hamsters exhibited islet function similar to non-diabetic controls, with significantly better results than diabetic controls. After two weeks, the islets were well vascularized, with minimal inflammation, and β-cells maintained adequate insulin production for at least 129 days [86].

Thus, the gastric submucosa is a promising site for pancreatic islet transplantation due to its good vascularization and endoscopic accessibility. Despite positive outcomes, such as improved glycemic control in animal models, the procedure’s effectiveness may be affected by the interaction between the graft and the submucosal environment and the use of extracellular matrices. Comparisons with other sites (e.g., the liver) suggest alternatives that may yield better results; however, the gastric submucosa still warrants further investigation to optimize the technique and identify the optimal anatomical site for implantation.

## 10. Bioartificial Pancreas Transplantation into the Intrapleural Space

Kaur et al. proposed the intrapleural space as a promising site for implanting tissue grafts with high metabolic activity, such as hepatocytes and pancreatic islets, due to its higher oxygen tension than in intraperitoneal spaces and other regions [87].

Lei et al. conducted a pilot study in which they transplanted pancreatic islets into the intrapleural space. Due to the relatively small donor size, the authors transplanted a dose of 15,500 IEQ per kilogram of the RBW, as prior studies had indicated that a dose of approximately 25,000 IEQ/kg/RBW was necessary to achieve robust insulin-free normoglycemia [88]. However, even with the reduced dose, euglycemia was maintained in vivo. The intravenous glucose tolerance test conducted 28 and 92 days post-transplantation showed robust glycemic control, similar to control animals. These findings suggest that the pleural space provides an effective environment for the graft and the maintenance of islet function. The observed efficiency may be attributed to three factors: (1) an optimized environment with higher oxygen partial pressure due to direct oxygen diffusion from the pulmonary parenchyma; (2) the reduced loss of graft tissue resulting from the absence of an instant blood-mediated inflammatory reaction; and (3) lower exposure of the graft to toxic levels of immunosuppressive agents [88].

Thus, the intrapleural space emerges as a promising site for islet transplantation, demonstrating effectiveness in maintaining euglycemia, even with a reduced graft dose. This is attributed to high oxygen partial pressure levels and a reduced inflammatory response. The minimally invasive approach, free from complications such as pneumothorax, hemothorax, and the formation of pulmonary adhesions, reinforces the procedure’s viability. Although complications are theoretical and rare, further studies are needed to confirm this technique’s long-term efficacy and safety for islet transplantation.

Given the challenges in identifying ideal sites for implanting the bioartificial pancreas, addressing T1DM requires an integrated approach combining bioengineering advancements, medical technology, and translational research. A thorough evaluation of potential sites, such as the liver, omentum, subcutaneous, and intrapleural space, and the development of supportive devices and biofunctionalized scaffolds pave the way for a promising future.

## 11. Assessing Biocompatible Scaffolds for Pancreatic Islet Cell Transplantation

Recently, natural polymers, including decellularized biological matrices, have been proposed for producing scaffolds. Biological scaffolds are biocompatible, natural mimickers of the extracellular matrix. They provide adequate immune protection, guarantee good vascularization, decrease the use of immunosuppressants, and create a surrounding ambiance for islet cells that effectively replicate the pancreas microenvironment. In addition, biological scaffolds are biodegradable, therefore possessing the necessary versatility, stiffness, porosity, gelation kinetics, and degradation properties to be cross-linked with other molecules [89]. Many extracellular matrix components can be utilized when preparing natural scaffolds for islet cell transplantation, including collagen, fibronectin, laminins, glycosaminoglycans, and fibrin. In this scenario, integrin receptors bind to and interact with the extracellular matrix components, contributing to the natural scaffold production, integrating each element, and mediating cellular responses downstream of the extracellular matrix engagement [90]. Many are the examples of biocompatible scaffolds for islet cell transplantation under diabetic conditions.

Hendrawan et al. [91] conducted a study using poly-l-lactide matrix to experimentally transplant allogeneic islet cell implants in streptozotocin-induced diabetic rats to treat T1DM. They proposed a three-dimensional proprietary poly-(l-lactide) matrix to induce islet cell implantation success and reduce the glycemic levels of diabetic rats. The results demonstrated that 82% of seeded islet cells attached to the matrices, with a significant reduction in glycemic parameters. Li et al. [92] also proposed a novel biocompatible scaffold for islet cell transplantation. They used microencapsulation to unite islet cells in an anti-adhesive core–shell microgel. Then, they used macroencapsulation with a vascularized hydrogel scaffold to enhance transplantation success. The hydrogel scaffold contained methacrylated gelatin (GelMA), vascular endothelial growth factor (VEGF), and methacrylated heparin (HepMA). This delivered VEGF and promoted angiogenesis. The islet-laden core–shell microgels utilized methacrylated hyaluronic acid (HAMA) and poly(ethylene glycol) diacrylate (PEGDA)/carboxybetaine methacrylate (CBMA) to promote a layer, providing a favorable microenvironment for the success of transplantation. As a result, the increased adherence and survival of the islet cell transplants were able to reduce glycemia and regularize the glycemic control of the treated animals. Finally, Sevastianov et al. [93] evaluated the effectiveness of a tissue-engineered construct based on a decellularized scaffold to induce normoglycemia in streptozotocin-induced diabetic animals. The results demonstrated that the fine-dispersed tissue-specific scaffold significantly induced the islet cell transplants’ more prolonged survival and efficient function. The treated animals had glycemic levels under control after transplantation.

## 12. Conclusions

Identifying ideal sites for bioartificial pancreas implantation is a multidisciplinary and multidimensional challenge that requires evaluating various factors, including vascularization, immune response, graft support, and surgical accessibility. Despite significant advancements in recent decades, no implantation site has been universally recognized as ideal for meeting the needs of patients with T1DM. Traditionally used sites, such as the liver, omentum, and renal subcapsular space, offer specific advantages and limitations. However, emerging alternatives, such as subcutaneous and intrapleural spaces, show promise when combined with bioengineering approaches. Figure 2 illustrates the implantation sites discussed within this article.

The choice of implantation site must balance the preservation of islet viability with the mitigation of complications such as exacerbated immune responses and hypoxia. Advanced technologies, such as vascularized support devices and biofunctionalized scaffolds, may improve transplant efficacy and prolong graft functionality. Further preclinical and clinical research will be critical for validating these alternatives and translating laboratory advancements into safe and effective clinical practices. This review contributes to developing innovative strategies for treating T1DM, bringing us closer to achieving more efficient and less invasive glycemic control for patients.

## Figures and Tables

**Figure 1 bioengineering-12-00499-f001:**
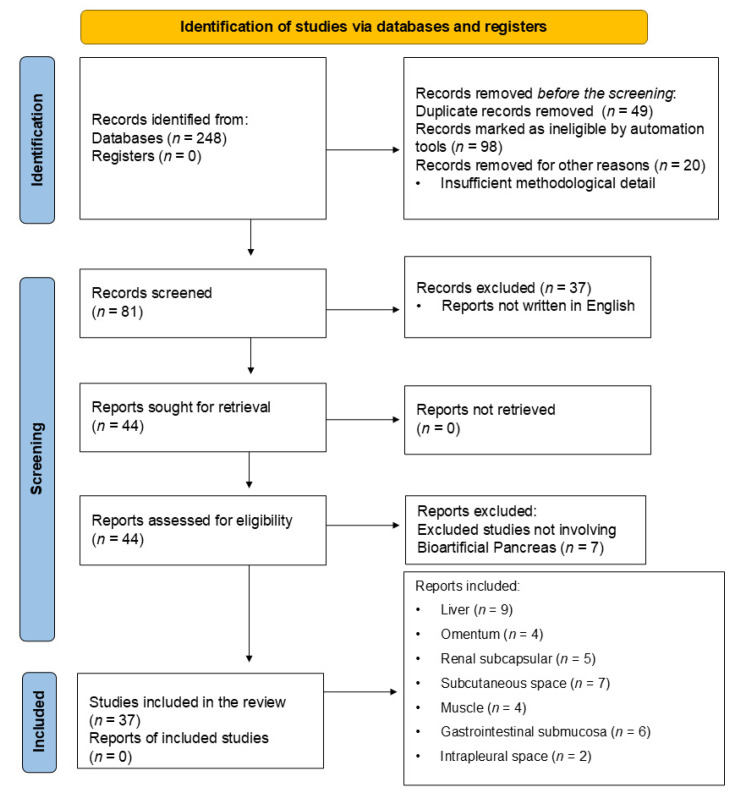
PRISMA flow diagram depicting the study selection process.

**Figure 2 bioengineering-12-00499-f002:**
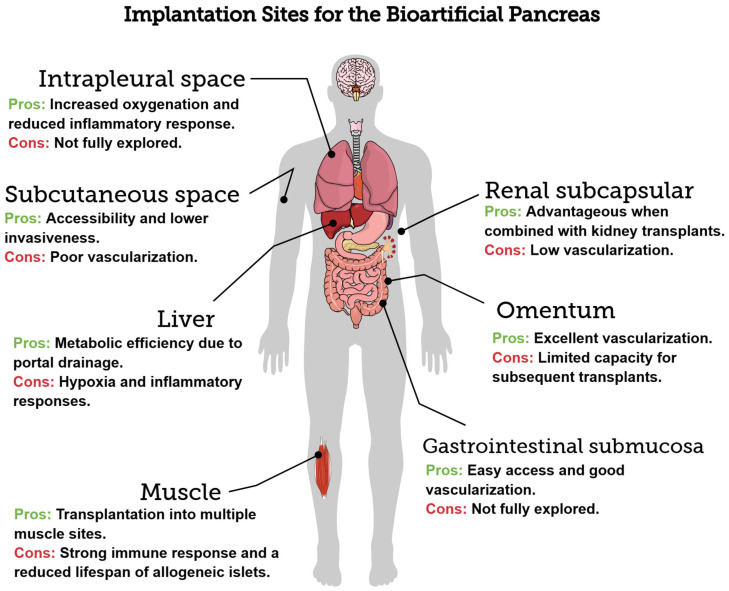
The implantation sites for the bioartificial pancreas are discussed in the above sections, created with Mind the Graph (https://mindthegraph.com/ (accessed on 24 April 2025)).

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
