# Peer review of "Assessing Implantation Sites for Pancreatic Islet Cell Transplantation: Implications for Type 1 Diabetes Mellitus Treatment"

_bioengineering, 2025, doi:10.3390/bioengineering12050499_

Round 1

Reviewer 1 Report

Comments and Suggestions for Authors

The paper is focused on Assessing Implantation Sites for the Bioartificial Pancreas and their Implications for Type 1 Diabetes Mellitus Treatment: A Review of the Current Literature and Future Prospects. The manuscript must be extensively revised and improved for having a chance. Few of my main concerns are listed below.

  1. The title is much too long and incorrect. Above the title is already written Review, so it sounds duplicating. Also, a Review implies always pointing out the prospects. Always the prospects are future considerations, so future never needed when it is about prospect. Remove “A Review of the Current Literature and Future Prospects”.
  2. 9 authors for a manuscript of 9 pages of main text, in MDPI format (which occupies 2/3 of a page) are too many. What they have done, as the manuscript is not at all in its best shape and content?
  3. Email of each author must be provided in affiliations section.
  4. iThenticate % of 25% is much too high, revise the entire manuscript regarding this aspect.
  5. The review has 1 table and NO figure. Very poor presentation.
  6. L90-95. A real aim of a study should be addressed from the point of view of the novelty/special aspects it brings to the field, or the reason for choosing the topic. Develop the aim of the study as better as you can. What differentiate your paper from other in the same topic? Give a reason for interest in this paper. The actual aim of the study gave no reason for this paper and no interest.
  7. How have you searched the literature? No Treemap or PRISMA flow chart is presented. The authors should be interested in having a good justification of their topic, and they will make a short search (graphically) related to the impact of the topic on the general literature (using i.e. Web of Science and PubMed data bases which the authors already considered); they can obtain a scientometric figure/Treemap chart which would be interesting in justifying (or not) the novelty/impact of the topic in the frame of the existing literature.
  8. Discuss which is the best location for islet cell transplantation in the cases of pancreatic cancer where total pancreatectomy is sometimes needed and liver metastasis can occur and describe the frame in this context. I suggest checking and referring to https://doi.org/10.3390/diagnostics10110869
  9. In chapter 4. Bioartificial Pancreas Transplantation into the Liver. Detail better the impact of hepatic microenvironment such as increased oxidative stress on the survival of transplanted beta cells as well as the impact of microenvironment modification in certain hepatic diseases on the function of these transplanted beta cells, considering https://doi.org/10.1016/j.biopha.2022.113238
  10. In the review, the notion bioartificial pancreas must be clearly defined as islet cell transplantation does not comprehensively cover this notion. The notion could also refer to electronic implantable devices that use insulin such as an implantable insulin pump. 

Author Response

RESPONSE TO REVIEWERS' COMMENTS

Manuscript number: bioengineering-3584982― Bioengineering (MDPI)

"Assessing Implantation Sites for the Pancreatic Islet Cell Transplantation: Implications for Type 1 Diabetes Mellitus Treatment"

The authors of this document wish to express their deepest gratitude to the Editor-in-Chief and the Reviewer for their thorough and insightful evaluation of our manuscript. Their expert feedback has been invaluable in enhancing the quality of our work. We have carefully considered and diligently implemented each suggestion, significantly improving the manuscript. We have made substantial revisions to address the points raised. These noteworthy changes are marked mainly with YELLOW-highlighted text throughout the document for ease of reference. A note will be provided for the referee's attention for corrections highlighted in a different color. Additionally, we have prepared a detailed and comprehensive response to each comment and suggestion. This response is organized in a "point-by-point" format below, ensuring that every concern has been thoroughly addressed and explained. We sincerely appreciate the time and effort invested by the Editor-in-Chief and the Reviewer, and we believe their contributions have significantly strengthened the final version of our manuscript.

REVIEWER #1

General comment

The paper is focused on Assessing Implantation Sites for the Bioartificial Pancreas and their Implications for Type 1 Diabetes Mellitus Treatment: A Review of the Current Literature and Future Prospects. The manuscript must be extensively revised and improved for having a chance. Few of my main concerns are listed below.

General response

Dear Erudite Reviewer, thank you for taking the time to revise our manuscript and for allowing us to improve based on your valuable comments and suggestions. After addressing all your comments and suggestions regarding our manuscript text, we are confident that a significantly enhanced manuscript version has emerged. We are excited to resubmit the modified version for your perusal and reevaluation. Thank you for your brilliant insights, essential contributions, and feedback. You do have an eye for improvement. As a gesture of our utmost respect for you, we would like to provide you with a detailed and comprehensive point-by-point response to your comments below. Thank you once again for your time and patience in revising our article.

Comment #1

The title is much too long and incorrect. Above the title is already written Review, so it sounds duplicating. Also, a Review implies always pointing out the prospects. Always the prospects are future considerations, so future never needed when it is about prospect. Remove “A Review of the Current Literature and Future Prospects”.

Response

Dear Erudite Reviewer, thank you for this comment. We agree that modifying the title to make it more accurate and legitimate would significantly enhance our manuscript’s quality and readability. Therefore, we made the necessary corrections, which are presented in Lines 2-3 on Page 1 of the revised manuscript document. The title is now offered as “Assessing Implantation Sites for the Pancreatic Islet Cell Transplantation: Implications for Type 1 Diabetes Mellitus Treatment. Thank you for your patience and guidance throughout this critical peer-review process.

Comment #2

9 authors for a manuscript of 9 pages of main text, in MDPI format (which occupies 2/3 of a page) are too many. What they have done, as the manuscript is not at all in its best shape and content?

Response

Dear Erudite Reviewer, we appreciate your concern and are thankful for the opportunity to explain our authorship list. Besides the fact that our manuscript has many authors, we also have many text sections. In its current form, our manuscript presents eleven sections divided into specific implantation sites for the bioartificial pancreas, the prospects, and the literature basis of our research. Our authorship is accurate since our authors participated in the manuscript’s writing, original draft preparation, writing, review, editing, and conceptualization. All authors have read and approved the submitted version to this critical journal, Bioengineering. Therefore, our authorship aligns with the International Committee of Medical Journal Editors (ICMJE) guidelines for authorship preparation and decision. In addition, during this round of revisions, our co-authors significantly contributed to performing the critical and essential suggestions and recommendations proposed by you and the other reviewers who assessed our manuscript’s suitability for publication.

            Finally, we understand that your first opinion of our manuscript might have been that it wasn’t in its best format. However, we were truly aware of the suggestions and recommendations you raised during your critical evaluation of our work. We have addressed your comments and suggestions, resulting in a significantly improved manuscript. We are excited to resubmit the revised version for your reevaluation. Thank you for your valuable insights and feedback.

Comment #3

Email of each author must be provided in affiliations section

Response

Dear Erudite Reviewer, thank you for these essential suggestions. To improve our manuscript accordingly, the email addresses of my co-authors have been included in the affiliations list. Their email addresses are in Lines 8-24 on Page 1 of our revised manuscript document. Thank you for your time and consideration!

Comment #4

iThenticate % of 25% is much too high, revise the entire manuscript regarding this aspect.

Response

Dear Erudite Reviewer, thank you for this comment. We understand your concern and are eager to anticipate your positive response. We have diligently revised the entire manuscript to avoid similarity. We can assure you that our similarity index has significantly decreased in this revised version of our manuscript. You can find our work in reducing the similarity index of our manuscript by searching for the PINK highlighted text in the entire manuscript. Many parts of the text have been modified, including Lines 131-133 on Page 3, Line 171 on Page 5, Lines 206-207 on Page 5, Lines 225-227 on Page 6, Lines 319-321 on Page 8, Lines 356-357 on Page 8, and Lines 406-407 on Page 9. According to our knowledge, similarity now only remains in the reference list and in the manuscript’s disclosures, which are identical for all MDPI’s articles and are not in our hands to be modified since they are standard for the journal.

Thank you for assessing our manuscript with the utmost transparency and criticism. We are confident that your recommendations significantly improved the quality of our text!

Comment #5

The review has 1 table and NO figure. Very poor presentation.

Response

Dear Erudite Reviewer, thank you for this comment. We agree that adding an illustrative figure to our manuscript would enhance its quality. Therefore, we included Figure 2 in our revised manuscript. This figure revises the main implantation sites for the bioartificial pancreas discussed in our manuscript. We used professional software to create this figure, since it was designed using Mind the Graph (https://mindthegraph.com/). The new Figure 2 is on Page 13 of the revised document. Additionally, you will find the figure’s first mention in Lines 550-551 on Page 12 and its legend in Lines 554-555 on Page 13.

            Again, thank you for your patience and guidance. Your recommendations were instrumental in improving our manuscript.

Comment #6

L90-95. A real aim of a study should be addressed from the point of view of the novelty/special aspects it brings to the field, or the reason for choosing the topic. Develop the aim of the study as better as you can. What differentiate your paper from other in the same topic? Give a reason for interest in this paper. The actual aim of the study gave no reason for this paper and no interest.

Response

Dear Erudite Reviewer, you are entirely correct, and we agree that highlighting our paper's novelty in the light of previously published articles would undoubtedly enhance our manuscript’s quality and make it more citable. Therefore, we implemented Lines 94-118 on Pages 2-3 in our comprehensive analysis. No review has comprehensively examined the complete literature on bioartificial pancreas implantation sites, especially given the numerous recent studies. For instance, Berney et al. focused on the biological aspects of the microenvironment's impact on implant success, rather than implantation sites. Zhu et al. reviewed potential sites in 2018 but did not cover all recent findings, omitting the intrapleural space and gastrointestinal submucosa. Hwang et al. also explored implantation sites but prioritized microencapsulation techniques, lacking a thorough analysis of the sites themselves. Our review aims to evaluate all potential sites for bioartificial pancreas transplantation, focusing on factors like proximity to blood supply, graft support capacity, and surgical feasibility. We will also explore the implications for treating Type 1 Diabetes Mellitus (T1DM), addressing a critical gap in previous research. Ultimately, the goal is to enhance the functionality and longevity of the bioartificial pancreas in T1DM patients and inform future clinical practices.

Comment #7

How have you searched the literature? No Treemap or PRISMA flow chart is presented. The authors should be interested in having a good justification of their topic, and they will make a short search (graphically) related to the impact of the topic on the general literature (using i.e. Web of Science and PubMed data bases which the authors already considered); they can obtain a scientometric figure/Treemap chart which would be interesting in justifying (or not) the novelty/impact of the topic in the frame of the existing literature.

Response

Dear Erudite Reviewer, thank you for this insightful comment and suggestion. To improve our manuscript accordingly, we included Figure 1 on Page 4 (its legend is in Line 154 on Page 4) of the revised manuscript document. Additionally, we updated Section 2’s title in Line 119 on Page 3 and subsections in Lines 120 and 138 on Page 3. Subsection 2.2 in Lines 138-152 on Pages 3-4 is wholly new and reports the selected studies' findings based on Figure 1, the PRISMA flowchart. This is a PRISMA flow diagram depicting our literature search process. This figure aims to provide a complete overview of the screening, identification, and selection of studies for our research. We strive to fully and correctly adhere to the utmost guidelines when reporting our findings. Thank you for highlighting the necessity of a scientometric figure to ensure our methodology could be understood entirely. Thank you for everything!

Comment #8

Discuss which is the best location for islet cell transplantation in the cases of pancreatic cancer where total pancreatectomy is sometimes needed and liver metastasis can occur and describe the frame in this context. I suggest checking and referring to https://doi.org/10.3390/diagnostics10110869

Response

Dear Erudite Reviewer, thank you for this insightful comment. We agree with you. Therefore, considering the aspects you raised, we implemented Lines 267-273 on Page 7. To maintain our utmost responsibility to respond to your comment accordingly, we cited your suggested reference alongside other references in our newly added text. The freshly added reference appears in Lines 712-173 on Page 17 of our reference list. The liver is a common alternative for islet cell transplantation due to its vascular access and ability to support insulin production through implants. However, in total pancreatectomy cases involving pancreatic cancer, it’s essential to consider the liver's potential for metastasis, which can affect its suitability for transplantation. If the liver is deemed unsuitable, alternative transplant sites must be explored. We highlighted this alternative throughout our entire manuscript.

            Again, thank you for assessing our manuscript. It is such an honor to communicate with such an esteemed reviewer. Thank you for everything!

Comment #9

In chapter 4. Bioartificial Pancreas Transplantation into the Liver. Detail better the impact of hepatic microenvironment such as increased oxidative stress on the survival of transplanted beta cells as well as the impact of microenvironment modification in certain hepatic diseases on the function of these transplanted beta cells, considering https://doi.org/10.1016/j.biopha.2022.113238

Response

Dear Erudite Reviewer, thank you for your insightful comment. We completely agree with your observations. In response to your points, we have made revisions in Lines 252-261 on Pages 6-7. To ensure we address your feedback thoroughly, we have included your suggested reference and additional sources in our newly added text. This reference can be found in Lines 694-696 on Page 16 of our reference list. Modifications to the liver microenvironment can threaten pancreatic cell survival in transplants, as oxidative stress is linked to many diseases, including liver issues. This stress alters the environment, leading to increased toxicity and apoptosis, which diminishes healthy cell survival. Supplementing islet preparations with mesenchymal stromal cells may enhance islet survival during isolation and transplantation, reducing stress and improving post-transplant functionality. Lowering oxidative stress during islet transplantation may also improve long-term outcomes for diabetic patients by enhancing transplant survival and restoring glucose homeostasis.

Thank you for reviewing our manuscript. It is an honor to communicate with such a respected reviewer, and we appreciate your insights!

Comment #10

In the review, the notion bioartificial pancreas must be clearly defined as islet cell transplantation does not comprehensively cover this notion. The notion could also refer to electronic implantable devices that use insulin such as an implantable insulin pump.

Response

Dear Erudite Reviewer, thank you for this comment and insightful suggestion. We agree that a bioartificial pancreas must encompass implantable electronic devices that “secrete” insulin under determined concentrations in response to glucose alterations. However, our manuscript delves exclusively into the rationale behind the best implantation sites for the bioartificial pancreas in terms of beta cell transplantation. We work solely with nutrient and oxygen supplies, bloodstream correlation to implantable sites, and many other aspects of beta cell transplantation, since our primary focus is not implantable electronic devices but rather islet cell transplantation. Because of this, we altered our title to “Assessing Implantation Sites for the Pancreatic Islet Cell Transplantation: Implications for Type 1 Diabetes Mellitus Treatment” in Lines 2-3 on Page 1 to strictly imply our focus, which is to assess the implantation sites for the pancreatic islet cell transplantation and their implications for type 1 diabetes mellitus treatment.

            We thank you for raising this concern and hope you can approve the modification to our title as significant to resolve this comment of yours. Thank you for everything!

I, the corresponding author of the manuscript "Assessing Implantation Sites for the Pancreatic Islet Cell Transplantation: Implications for Type 1 Diabetes Mellitus Treatment" under the assigned ID bioengineering-3584982, on behalf of my coauthors, once again extend my heartfelt gratitude to the knowledgeable Editor-in-Chief and reviewers for their time and expertise in revising our manuscript. After we addressed their constructive and refined feedback and suggestions, a significantly improved manuscript version emerged. Undoubtedly, their insightful suggestions and feedback have significantly enhanced the quality of our manuscript. We respectfully are at the disposal of the Editor-in-Chief and the Reviewer to address any additional suggestions regarding our publication. If you are satisfied with our newly refined and significantly improved version, we look forward to the acceptance of our article for publication in this prestigious journal, Bioengineering. Thank you once again for your time and expertise.

Reviewer 2 Report

Comments and Suggestions for Authors

The review critically evaluates the potential implantation sites of a bioartificial pancreas for the treatment of Type 1 diabetes. The transplantation of pancreatic islets can be viewed as a less invasive and, consequently, safer alternative to whole pancreas transplantation. The survival rate of islets is approximately 80% five years post-transplant, with full insulin independence observed in about 50% of patients during the same period. To enhance transplant survival rates, it is essential to provide islets with an optimal microenvironment, which is largely determined by the site of implantation. The review encompasses the advantages and disadvantages of various methods for administering the bioartificial pancreas, including locations such as the liver, omentum, renal subcutaneous space, subcutaneous tissue, gastrointestinal submucosa, and intrapleural space.  There are no fundamental comments.

Nevertheless, the review contains some issues to be addressed.

  1. A flowchart illustrating the selection process for the articles should be included, detailing the number of studies that met the inclusion criteria for the review. Additionally, the date of the literature search should be specified.
  2. Tissue engineering methodologies are utilized in the development of the bioartificial pancreas. The authors should include the keywords “pancreas tissue engineering” as part of their search strategy.
  3. It would be useful for readers to review the results of a literature search, including the number of clinical and experimental studies, as well as articles discussing various islet implantation sites. This data should be added to the article, possibly in the form of tables, figures or diagrams.
  4. Various scaffolds, such as poly-L-lactide scaffolds (Hendrawan, Siufui et al. “Allogeneic islet cells implant on poly-L-lactide matrix to reduce hyperglycemia in streptozotocin-induced diabetic rat.” Pancreatology: official journal of the International Association of Pancreatology (IAP), vol. 17, no. 3, 2017, pp. 411-418. doi:10.1016/j.pan.2017.02.017), hydrogel scaffolds containing methacrylated gelatin and methacrylated heparin (Haofei et al. “A novel bioartificial pancreas fabricated via islets microencapsulation in anti-adhesive core-shell microgels and macroencapsulation in a hydrogel scaffold prevascularized in vivo.” Bioactive Materials, vol. 27, pp. 362-376, April 20, 2023, doi:10.1016/j.bioactmat.2023.04.011), and decellularized scaffolds (Sevastianov, Victor I et al. “A Tissue-Engineered Construct Based on a Decellularized Scaffold and the Islets of Langerhans: A Streptozotocin-Induced Diabetic Model.” Life (Basel, Switzerland), vol. 14, no. 11, 1505, November 19, 2024, doi:10.3390/life14111505) have been utilized in experimental in vivo studies of the artificial pancreas. The authors should consider including a separate section dedicated to scaffolds for pancreatic tissue engineering.
  5. It is also recommended to increase the number of actual examples in Table 1.

Author Response

RESPONSE TO REVIEWERS' COMMENTS

Manuscript number: bioengineering-3584982― Bioengineering (MDPI)

"Assessing Implantation Sites for the Pancreatic Islet Cell Transplantation: Implications for Type 1 Diabetes Mellitus Treatment"

The authors of this document wish to express their deepest gratitude to the Editor-in-Chief and the Reviewer for their thorough and insightful evaluation of our manuscript. Their expert feedback has been invaluable in enhancing the quality of our work. We have carefully considered and diligently implemented each suggestion, significantly improving the manuscript. We have made substantial revisions to address the points raised. These noteworthy changes are marked mainly with YELLOW-highlighted text throughout the document for ease of reference. A note will be provided for the referee's attention for corrections highlighted in a different color. Additionally, we have prepared a detailed and comprehensive response to each comment and suggestion. This response is organized in a "point-by-point" format below, ensuring that every concern has been thoroughly addressed and explained. We sincerely appreciate the time and effort invested by the Editor-in-Chief and the Reviewer, and we believe their contributions have significantly strengthened the final version of our manuscript.

REVIEWER #2

General comment

The review critically evaluates the potential implantation sites of a bioartificial pancreas for the treatment of Type 1 diabetes. The transplantation of pancreatic islets can be viewed as a less invasive and, consequently, safer alternative to whole pancreas transplantation. The survival rate of islets is approximately 80% five years post-transplant, with full insulin independence observed in about 50% of patients during the same period. To enhance transplant survival rates, it is essential to provide islets with an optimal microenvironment, which is largely determined by the site of implantation. The review encompasses the advantages and disadvantages of various methods for administering the bioartificial pancreas, including locations such as the liver, omentum, renal subcutaneous space, subcutaneous tissue, gastrointestinal submucosa, and intrapleural space.  There are no fundamental comments. Nevertheless, the review contains some issues to be addressed.

General response

Dear Erudite Reviewer, thank you for taking the time to revise our manuscript and for allowing us to improve based on your valuable comments and suggestions. After addressing all your comments and suggestions regarding our manuscript text, we are confident that a significantly enhanced manuscript version has emerged. We are excited to resubmit the modified version for your perusal and reevaluation. Thank you for your brilliant insights, essential contributions, and feedback. You do have an eye for improvement. As a gesture of our utmost respect for you, we would like to provide you with a detailed and comprehensive point-by-point response to your comments below. Thank you once again for your time and patience in revising our article.

Comment #1

A flowchart illustrating the selection process for the articles should be included, detailing the number of studies that met the inclusion criteria for the review. Additionally, the date of the literature search should be specified.

Response

Dear Erudite Reviewer, thank you for your insightful comments and suggestions. To enhance our manuscript, we have included Figure 1 on Page 4, with its legend in Line 154 on the same page. We also updated the title of Section 2 in Line 119 and revised the subsections in Lines 120 and 138 on Page 3. Additionally, Subsection 2.2, found in Lines 139-152 on Pages 3-4, is entirely new and presents the findings from the selected studies, as outlined in Figure 1, which is the PRISMA flowchart. This figure illustrates our literature search process and provides a comprehensive overview of the screening, identification, and selection of studies for our research. To resolve your comment, we included the literature search date in Lines 136-137 on Page 3.

We are committed to reporting our findings according to the highest guidelines. Thank you for emphasizing the importance of including a scientometric figure to enhance the clarity of our methodology. We appreciate your support!

Comment #2

Tissue engineering methodologies are utilized in the development of the bioartificial pancreas. The authors should include the keywords “pancreas tissue engineering” as part of their search strategy.

Response

Dear Erudite Reviewer, thank you for this insightful comment. We agree with you. Therefore, we added the keyword to our methodology. Please find the inclusion in Line 123 on Page 3. Thank you for addressing critical suggestions and recommendations with us.

Comment #3

It would be useful for readers to review the results of a literature search, including the number of clinical and experimental studies, as well as articles discussing various islet implantation sites. This data should be added to the article, possibly in the form of tables, figures or diagrams.

Response

Dear Erudite Reviewer, Thank you for this comment. We completely agree with you, so we included Figure 2 on Page 13. This figure reviews the implantation sites for the bioartificial pancreas transplants and includes the pros and cons of each site. We believe this addition will greatly enhance the quality and readability of our manuscript since Figure 2 depicts a summary of our findings. We hope you can approve this additional content. Figure 2’s legend is disposed in Lines 554-555 on Page 13. The figure’s first mention is in Lines 550-551 on Page 13.

            Thank you for your time and patience in addressing critical comments with us! Your contributions have been invaluable in improving the quality of our text. Thank you for everything!

Comment #4

Various scaffolds, such as poly-L-lactide scaffolds (Hendrawan, Siufui et al. “Allogeneic islet cells implant on poly-L-lactide matrix to reduce hyperglycemia in streptozotocin-induced diabetic rat.” Pancreatology: official journal of the International Association of Pancreatology (IAP), vol. 17, no. 3, 2017, pp. 411-418. doi:10.1016/j.pan.2017.02.017), hydrogel scaffolds containing methacrylated gelatin and methacrylated heparin (Haofei et al. “A novel bioartificial pancreas fabricated via islets microencapsulation in anti-adhesive core-shell microgels and macroencapsulation in a hydrogel scaffold prevascularized in vivo.” Bioactive Materials, vol. 27, pp. 362-376, April 20, 2023, doi:10.1016/j.bioactmat.2023.04.011), and decellularized scaffolds (Sevastianov, Victor I et al. “A Tissue-Engineered Construct Based on a Decellularized Scaffold and the Islets of Langerhans: A Streptozotocin-Induced Diabetic Model.” Life (Basel, Switzerland), vol. 14, no. 11, 1505, November 19, 2024, doi:10.3390/life14111505) have been utilized in experimental in vivo studies of the artificial pancreas. The authors should consider including a separate section dedicated to scaffolds for pancreatic tissue engineering.

Response

Dear Erudite Reviewer, thank you for this recommendation. We agree that adding information regarding biocompatible scaffolds for pancreatic islet cell transplantation to our manuscript would undoubtedly enhance our manuscript’s quality and readability. Therefore, we assessed this information by including Lines 504-540 on Page 12. A new Section 11 was created in Lines 504-505 on Page 12. We cited every reference you provided, which can be found in the reference list in Lines 810-819 on Page 19, alongside additional references. We believe our manuscript has significantly improved after addressing your critical recommendations. Thank you for your time and consideration!

Comment #5

It is also recommended to increase the number of actual examples in Table 1.

Response

Dear Erudite Reviewer, thank you for this comment. We agree that the previous Table 1 was limited to advancing the knowledge we would like to advance in our manuscript. Therefore, we resolved to exclude it from the manuscript. In return, we included Figure 2 on Page 13. Its first mention is presented in Lines 550-551 on Page 12, and its legend is in Lines 554-555 on Page 13. This figure perfectly substitutes Table 1 because Figure 2 is a complete summary of our findings. The main objective of our manuscript was to summarize the possible implantation sites for islet cell transplantation, which Figure 2 depicts with mastery, highlighting the pros and cons of each site. In addition, we believe our manuscript did not necessitate more text, but illustrations, which we have done with the addition of Figure 2.

            We hope you can approve our corrections. We are thankful for the opportunity to communicate with such an esteemed reviewer. Your critical comments have been instrumental in reshaping our manuscript for the better.

I, the corresponding author of the manuscript "Assessing Implantation Sites for the Pancreatic Islet Cell Transplantation: Implications for Type 1 Diabetes Mellitus Treatment" under the assigned ID bioengineering-3584982, on behalf of my coauthors, once again extend my heartfelt gratitude to the knowledgeable Editor-in-Chief and reviewers for their time and expertise in revising our manuscript. After we addressed their constructive and refined feedback and suggestions, a significantly improved manuscript version emerged. Undoubtedly, their insightful suggestions and feedback have significantly enhanced the quality of our manuscript. We respectfully are at the disposal of the Editor-in-Chief and the Reviewer to address any additional suggestions regarding our publication. If you are satisfied with our newly refined and significantly improved version, we look forward to the acceptance of our article for publication in this prestigious journal, Bioengineering. Thank you once again for your time and expertise.

Round 2

Reviewer 1 Report

Comments and Suggestions for Authors

The authors have significantly improved the manuscript based on the recommendations received.